# Benchmarking reveals superiority of deep learning variant callers on bacterial nanopore sequence data

**Michael B Hall**[1]*, **Ryan R Wick**[1,2], **Louise M Judd**[1,2], **An N Nguyen**[1], **Eike J Steinig**[1], **Ouli Xie**[3,4], **Mark Davies**[1], **Torsten Seemann**[1,2], **Timothy P Stinear**[1,2]†, **Lachlan Coin**[1]†

[1]Department of Microbiology and Immunology, The University of Melbourne, at the Peter Doherty Institute for Infection and Immunity, Melbourne, Australia; [2]Centre for Pathogen Genomics, The University of Melbourne, Melbourne, Australia; [3]Department of Infectious Diseases, The University of Melbourne, at the Peter Doherty Institute for Infection and Immunity, Melbourne, Australia; [4]Monash Infectious Diseases, Monash Health, Melbourne, Australia

**Abstract** Variant calling is fundamental in bacterial genomics, underpinning the identification of disease transmission clusters, the construction of phylogenetic trees, and antimicrobial resistance detection. This study presents a comprehensive benchmarking of variant calling accuracy in bacterial genomes using Oxford Nanopore Technologies (ONT) sequencing data. We evaluated three ONT basecalling models and both simplex (single-strand) and duplex (dual-strand) read types across 14 diverse bacterial species. Our findings reveal that deep learning-based variant callers, particularly Clair3 and DeepVariant, significantly outperform traditional methods and even exceed the accuracy of Illumina sequencing, especially when applied to ONT's super-high accuracy model. ONT's superior performance is attributed to its ability to overcome Illumina's errors, which often arise from difficulties in aligning reads in repetitive and variant-dense genomic regions. Moreover, the use of high-performing variant callers with ONT's super-high accuracy data mitigates ONT's traditional errors in homopolymers. We also investigated the impact of read depth on variant calling, demonstrating that 10× depth of ONT super-accuracy data can achieve precision and recall comparable to, or better than, full-depth Illumina sequencing. These results underscore the potential of ONT sequencing, combined with advanced variant calling algorithms, to replace traditional short-read sequencing methods in bacterial genomics, particularly in resource-limited settings.

*For correspondence:
michael.hall2@unimelb.edu.au

†These authors contributed equally to this work

## eLife assessment

This **important** study shows how a combination of the latest generation of Oxford Nanopore Technology long reads with state-of-the art variant callers enables bacterial variant discovery at an accuracy that matches or exceeds the current "gold standard" with short reads. The work thus heralds a new era, in which Illumina short-read sequencing no longer rules supreme. While the inclusion of a larger number of reference genomes would have enabled an even more fine-grained analysis, the evidence as it is supports the claims of the authors **convincingly**. The work will be of interest to anyone performing sequencing for outbreak investigations, bacterial epidemiology, or similar studies.

**eLife digest** Imagine being part of a public health institution when, suddenly, cases of *Salmonella* surge across your country. You are facing an outbreak of this foodborne disease, and the clock is ticking. People are consuming a contaminated product that is making them sick; how do you identify related cases, track the source of the infection, and shut down its production?

In situations like these, scientists need to tell apart even closely related strains of the same bacterial species. This process, known as variant calling, relies on first analysing (or 'sequencing') the genetic information obtained from the bacteria of interest, then comparing it to a reference genome.

Currently, two main approaches are available for genome sequencing. Traditional 'short-read' technologies tend to be more accurate but less reliable when covering certain types of genomic regions. New 'long-read' approaches can bypass these limitations though they have historically been less accurate.

Comparison with a reference genome can be performed using a tool known as a variant caller. Many of the most effective ones are now based on artificial intelligence approaches such as deep learning. However, these have primarily been applied to human genomic data so far; it therefore remains unclear whether they are equally useful for bacterial genomes.

In response, Hall et al. set out to investigate the accuracy of four deep learning-based and three traditional variant callers on datasets from 14 bacterial species obtained via long-read approaches. Their respective performance was also benchmarked against a more conventional approach representing a standard of accuracy (that is, a popular, non-deep learning variant caller used on short-read datasets). These analyses were performed on a 'truthset' established by Hall et al., a collection of validated data that allowed them to assess the performance of the various tools tested.

The results show that, in this context, the deep learning variant callers more accurately detected genetic variations compared to the traditional approach. These tools, which could be run on standard laptops, were effective even with low amounts of sequencing data – making them useful even in settings where resources are limited.

Variant calling is an essential step in tracking the emergence and spread of disease, identifying new strains of bacteria, and examining their evolution. The findings by Hall et al. should therefore benefit various sectors, particularly clinical and public health laboratories.

## Introduction

Variant calling is a cornerstone of bacterial genomics as well as one of the major applications of next-generation sequencing. Its downstream applications include identification of disease transmission clusters, prediction of antimicrobial resistance, and phylogenetic tree construction and subsequent evolutionary analyses, to name a few (*Stimson et al., 2019*; *Sheka et al., 2021*; *Walker et al., 2022*; *Bertels et al., 2014*). Variant calling is used extensively in public health laboratories to inform decisions on managing bacterial outbreaks (*Gorrie et al., 2021*) and in molecular diagnostic laboratories as the basis for clinical decisions on how to best treat patients with disease (*Sherry et al., 2023*).

Over the last 15 years, short-read sequencing technologies, such as Illumina, have been the mainstay of variant calling in bacterial genomes, largely due to their relatively high level of basecalling accuracy. However, nanopore sequencing on devices from Oxford Nanopore Technologies (ONT) have emerged as an alternative technology. One of the major advantages of ONT sequencing from an infectious disease's public health perspective is the ability to generate sequencing data in near real time, as well as the portability of the devices, which has enabled researchers to sequence in remote regions, closer to the source of the disease outbreak (*Faria et al., 2016*; *Hoenen et al., 2016*). Limitations in ONT basecalling accuracy have historically limited its widespread adoption for bacterial genome variant calling (*Delahaye and Nicolas, 2021*). ONT have recently released a new R10.4.1 pore, along with a new basecaller (*Oxford Nanopore Technologies, 2023a*) with three different accuracy modes (fast, high-accuracy [hac], and super-accuracy [sup]). The basecaller also has the ability to identify a subset of paired reads for which both strands have been sequenced (duplex), leading to impressive gains in basecalling accuracy (*Sanderson et al., 2023*; *Sereika et al., 2022*).

A number of variant callers have been developed for ONT sequencing (*Edge and Bansal, 2019*; *Zheng et al., 2022*; *Ahsan et al., 2021*). However, to date, benchmarking studies have focused on

human genome variant calling, and have mostly used the older pores, which do not have the ability to identify duplex reads (*Olson et al., 2023*; *Olson et al., 2022*; *Pei et al., 2021*). In addition, modern deep learning-based variant callers use models trained on human DNA sequence only, leaving an open question of their generalisability to bacteria (*Zheng et al., 2022*; *Poplin et al., 2018*; *Ahsan et al., 2021*). Human genomes have a very different distribution of *k*-mers (segments of DNA sequence of length *k*) and patterns of DNA modification, and as such, results from human studies may not directly carry over into bacterial genomics. Moreover, there is substantial *k*-mer and DNA modification variation within bacteria, mandating a broad multi-species approach for evaluation (*Tourancheau et al., 2021*). Existing benchmarks for bacterial genomes, while immensely beneficial and thorough, only assess short-read Illumina data (*Bush, 2021*; *Bush et al., 2020*).

In this study, we conduct a benchmark of single nucleotide polymorphism (SNP) and insertion/deletion (indel) variant calling using ONT and Illumina sequencing across a comprehensive spectrum of 14 Gram-positive and Gram-negative bacterial species. We used the same DNA extractions for both Illumina and ONT sequencing to ensure our results are not biased by acquisition of new mutations during culture. We develop a novel strategy for generating benchmark variant truthsets in which we project variations from different strains onto our gold standard reference genomes in order to create biologically realistic distribution of SNPs and indels. We assess both deep learning-based and traditional variant calling methods and investigate the sources of errors and the impact of read depth on variant accuracy.

## Results
### Genome and variant truthset
Ground truth reference assemblies were generated for each sample using ONT and Illumina reads (see Genome assembly).

Creating a variant truthset for benchmarking is challenging (*Majidian et al., 2023*; *Li, 2014*). Calling variants against a sample's own reference yields no variants, so we generated a mutated reference. Instead of random mutations, we used a pseudo-real approach, applying real variants from a donor genome to the sample's reference (*Li, 2014*; *Li et al., 2018*). This approach has the advantage of a simulation, in that we can be certain of the truthset of variants, but with the added benefit of the variants being real differences between two genomes.

**Table 1.** Summary of the average nucleotide identity (ANI) and number of variants found between each sample and its donor genome.

| Sample | Species | ANI (%) | GC (%) | SNPs | Insertions | Deletions | Total variants |
|---|---|---|---|---|---|---|---|
| ATCC_33560__202309 | *Campylobacter jejuni* | 99.50 | 30.22 | 6369 | 117 | 106 | 6592 |
| ATCC_35221__202309 | *Campylobacter lari* | 98.64 | 29.81 | 16541 | 57 | 67 | 16665 |
| ATCC_25922__202309 | *Escherichia coli* | 99.50 | 50.42 | 4531 | 119 | 242 | 4892 |
| KPC2__202310 | *Klebsiella pneumoniae* | 99.50 | 57.15 | 15877 | 90 | 78 | 16045 |
| AJ292__202310 | *Klebsiella variicola* | 99.50 | 57.62 | 22850 | 95 | 98 | 23043 |
| ATCC_19119__202309 | *Listeria ivanovii* | 99.46 | 37.13 | 8451 | 187 | 259 | 8897 |
| ATCC_BAA-679__202309 | *Listeria monocytogenes* | 99.50 | 37.98 | 9090 | 66 | 78 | 9234 |
| ATCC_35897__202309 | *Listeria welshimeri* | 99.03 | 36.35 | 16953 | 130 | 133 | 17216 |
| AMtb_1__202402 | *Mycobacterium tuberculosis* | 99.73 | 65.62 | 2102 | 95 | 84 | 2281 |
| ATCC_10708__202309 | *Salmonella enterica* | 99.36 | 52.20 | 18784 | 210 | 189 | 19183 |
| BPH2947__202310 | *Staphylococcus aureus* | 99.48 | 32.80 | 7894 | 95 | 63 | 8052 |
| MMC234__202311 | *Streptococcus dysgalactiae* | 99.16 | 39.49 | 10474 | 82 | 100 | 10656 |
| RDH275__202311 | *Streptococcus pyogenes* | 99.50 | 38.32 | 5361 | 60 | 68 | 5489 |
| ATCC_17802__202309 | *Vibrio parahaemolyticus* | 98.75 | 45.32 | 57887 | 280 | 304 | 58471 |

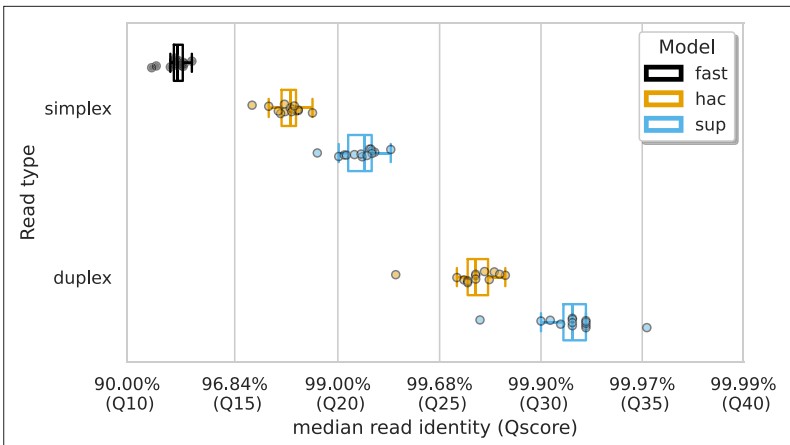

**Figure 1.** Median alignment-based read identity (x-axis) for each sample (points) stratified by basecalling model (colours) and read type (y-axis). The Qscore is the logarithmic transformation of the read identity, $Q = -10 \log_{10} P$, where $P$ is the read identity.

For each sample, we selected a donor genome with average nucleotide identity (ANI; a measure of similarity between two genomes) closest to 99.5% (see Truthset and reference generation). We identified all variants between the sample and donor using minimap2 (*Li, 2018*) and mummer (*Marçais et al., 2018*), intersected the variant sets, and removed overlaps and indels longer than 50 bp. This variant truthset was then applied to the sample's reference to create a mutated reference, ensuring no complications from large structural differences. While incorporating structural variation would be an interesting and useful addition to the current work, we chose to focus here on small (<50 bp) variants.

*Table 1* summarises the samples used, the number of variants, and the ANI between each sample and its donor. We analysed 14 samples from different species, spanning a wide range of GC content (30–66%). Despite the variation in SNP counts (2102-57887), the number of indels was consistent across samples (see *Supplementary file 1b* for details).

## Data quality

We analysed ONT data basecalled with three different accuracy models – fast, high accuracy (hac), and super-accuracy (sup) – along with different read types – simplex and duplex (see Basecalling and quality control). Duplex reads are those in which both DNA strands from a single molecule are sequenced back-to-back and basecalled together, whereas simplex reads are basecalled only using a single DNA strand. The median, unfiltered read identities, calculated by aligning reads to their respective assembly, are shown in *Figure 1*. Duplex reads basecalled with the sup model had the highest median read identity of 99.93% (Q32). The Qscore is the logarithmic transformation of the read identity, $Q = -10 \log_{10} P$, where $P$ is the read identity. This was followed by duplex hac (99.79% [Q27]), simplex sup (99.26% [Q21]), simplex hac (98.31% [Q18]), and simplex fast (94.09% [Q12]). Full summary statistics of the reads can be found in *Supplementary file 1a*.

## Which method is the best?

For this study, we benchmarked the performance of seven variant callers on ONT sequencing data: BCFtools (v1.19, *Danecek et al., 2021*), Clair3 (v1.0.5, *Zheng et al., 2022*), DeepVariant (v1.6.0, *Poplin et al., 2018*), FreeBayes (v1.3.7, *Garrison, 2012*), Longshot (v0.4.5, *Edge and Bansal, 2019*), Medaka(v1.11.3, *Oxford Nanopore Technologies, 2023a*; *Oxford Nanopore Technologies, 2023c*), and NanoCaller (v3.4.1, *Ahsan et al., 2021*). In addition, we called variants from each sample's Illumina data using Snippy (v4.6.0, *Seemann, 2015*) to act as a performance comparison.

Alignments of ONT reads to each sample's mutated reference (see Genome and variant truthset) were generated with minimap2 and provided to each variant caller (except Medaka, which takes reads directly). Variant calls were assessed against the truthset using vcfdist (v2.3.3, *Dunn and Narayanasamy, 2023*), classifying each variant as true positive (TP), false positive (FP), or false negative (FN). Precision, recall, and the F1 score were calculated for SNPs and indels at each VCF quality score

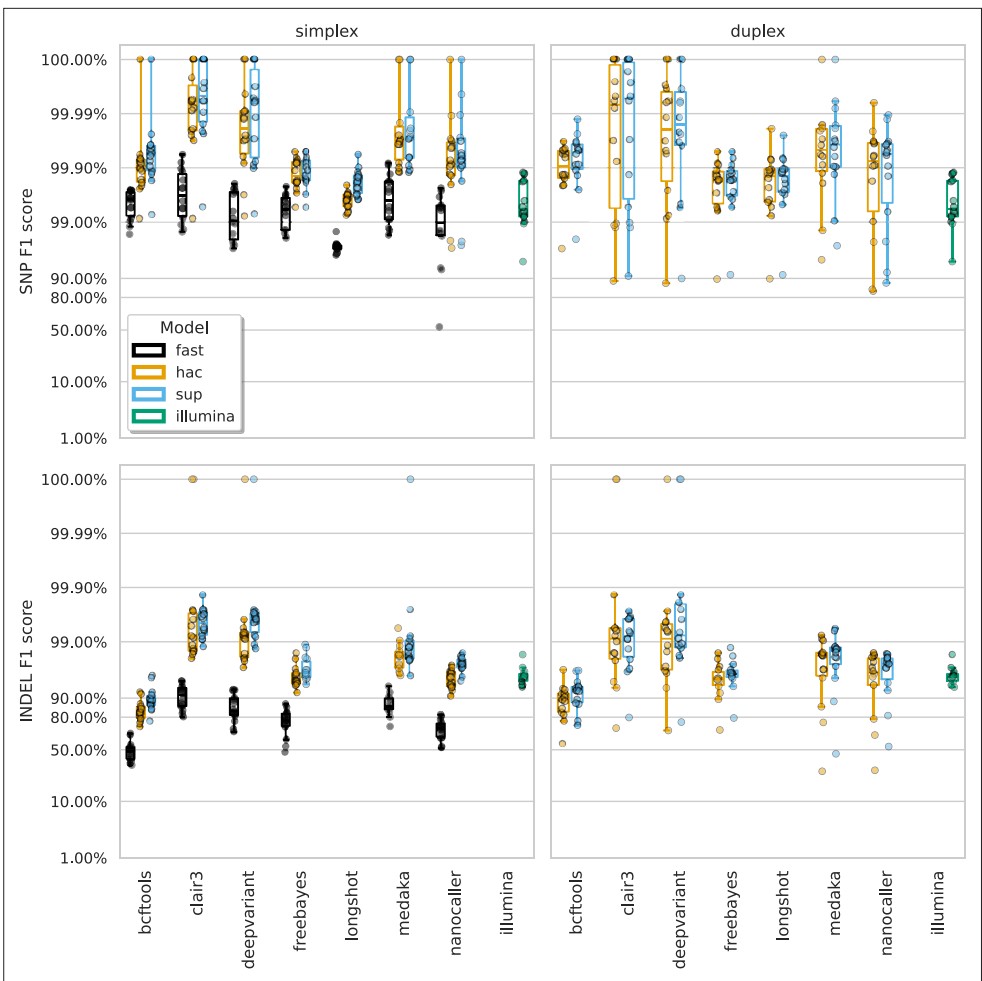

**Figure 2.** The highest F1 score for each sample (points), stratified by basecalling model (colours), variant type (rows), and read type (columns). Illumina results (green) are included as a reference and do not have different basecalling models or read types. Note, Longshot does not provide indel calls.

The online version of this article includes the following figure supplement(s) for figure 2:

**Figure supplement 1.** Precision at the highest F1 score for each sample (points), stratified by basecalling model (colours), variant type (rows), and read type (columns).

**Figure supplement 2.** Recall at the highest F1 score for each sample (points), stratified by basecalling model (colours), variant type (rows), and read type (columns).

**Figure supplement 3.** Clair3 sup model F1 score (y-axis) at the highest F1 score for each sample (x-axis), stratified by variant type (rows), and read type (shapes).

**Figure supplement 4.** Clair3 sup model precision (y-axis) at the highest F1 score for each sample (x-axis), stratified by variant type (rows), and read type (shapes).

**Figure supplement 5.** Clair3 sup model recall (y-axis) at the highest F1 score for each sample (x-axis), stratified by variant type (rows), and read type (shapes).

increment. *Figure 2* displays the highest F1 scores for each variant caller across samples, basecalling models, read types, and variant types.

The F1 score is the harmonic mean of precision and recall and acts as a good metric for overall evaluation. From *Figure 2* we see that Clair3 and DeepVariant produce the highest F1 scores for both SNPs and indels with both read types. Unsurprisingly, the sup basecalling model leads to the highest F1 scores across all variant callers, though hac is not much lower. SNP F1 scores of 99.99% are obtained from Clair3 and DeepVariant on sup-basecalled data. For indel calls, Clair3 achieves F1 scores of 99.53% and 99.20% for sup simplex and duplex, respectively, while DeepVariant scores

99.61% and 99.22%. The higher depth of the simplex reads likely explains why the best duplex indel F1 scores are slightly lower than simplex (see How much read depth is enough?). The precision and recall values at the highest F1 score can be seen in *Figure 2—figure supplement 1* and *Figure 2—figure supplement 2* (see *Supplementary file 1c* for a summary and *Supplementary file 1d* for full details) as well as results broken down by species for Clair3 with the sup model in *Figure 2—figure*

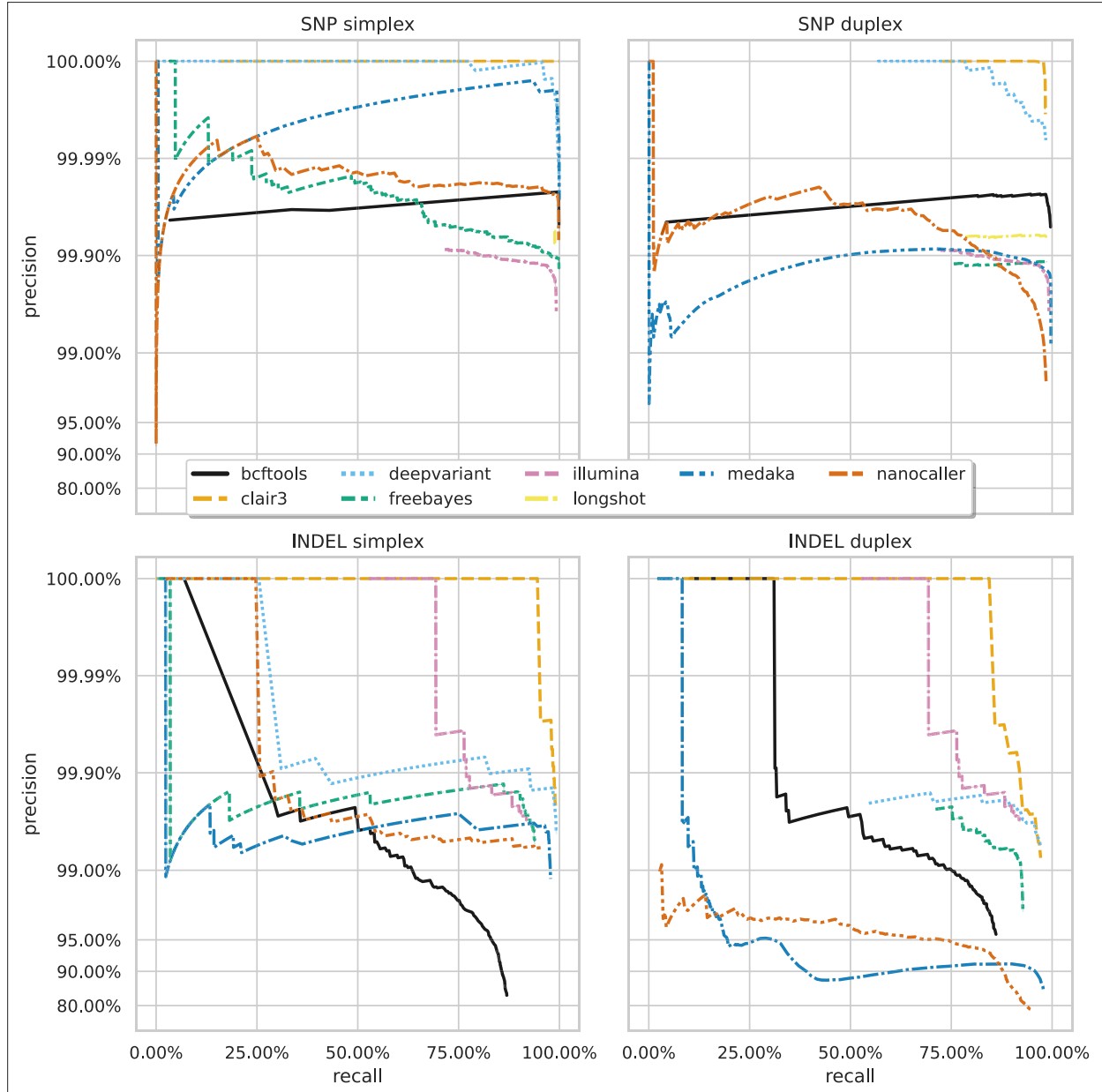

**Figure 3.** Precision and recall curves for each variant caller (colours and line styles) on sequencing data basecalled with the sup model, stratified by variant type (rows) and read type (columns) and aggregated across samples. The curves are generated by using increasing variant quality score thresholds to filter variants and calculating precision and recall at each threshold. The lowest threshold is the lower right part of the curve, moving to the highest at the top left. Note, Longshot does not provide indel calls.

The online version of this article includes the following figure supplement(s) for figure 3:

**Figure supplement 1.** Precision and recall curves for each variant caller (colours and line styles) on sequencing data basecalled with the hac model, stratified by variant type (rows) and ready type (columns).

**Figure supplement 2.** Precision and recall curves for each variant caller (colours and line styles) on sequencing data basecalled with the fast model, stratified by variant type (rows).

*supplement 3*, *Figure 2—figure supplement 4*, and *Figure 2—figure supplement 5*. Reads base-called with the fast model are an order of magnitude worse than the hac and sup models.

*Figure 3* shows the precision-recall curves for the sup basecalling model (see *Figure 3—figure supplement 1* and *Figure 3—figure supplement 2* for the hac and fast model curves, respectively) for each variant and read type – aggregated across samples to produce a single curve for each variant caller. Due to the right-angle-like shape of the Clair3 and DeepVariant curves, filtering based on low-value variant quality improves precision considerably for variant calls, without losing much recall. A similar pattern holds true for BCFtools SNP calls. The best Clair3 and DeepVariant F1 scores are obtained with no quality filtering on sup data, except for indels from duplex data where a quality filter of 4 provides the best F1. See *Supplementary file 1e* for the full details.

A striking feature of *Figure 2* and *Figure 3* is the comparison of deep learning-based variant callers (Clair3, DeepVariant, Medaka, and NanoCaller) to Illumina. For all variant and read types with hac or sup data, these deep learning methods match or surpass Illumina, with median best SNP and indel F1 scores of 99.45% and 95.76% for Illumina. Clair3 and DeepVariant, in particular, perform an order of magnitude better. Traditional variant callers (Longshot, BCFtools, and FreeBayes) match or slightly exceed Illumina for SNP calls with hac and sup data. FreeBayes matches Illumina for indel calls, but BCFtools shows reduced indel accuracy across all models and read types. Fast model ONT data has a lower F1 score than Illumina, only achieving parity in the best case for SNPs.

## Understanding missed and false calls

Conventional wisdom may leave readers surprised at finding that ONT data can provide better variant calls than Illumina. In order to convince ourselves (and others) of these results, we investigate the main causes for this difference.

Given the ONT read-level accuracy now exceeding Q20 (*Figure 1*; simplex sup), read length remains the primary difference between the two technologies. *Figure 2—figure supplement 1* shows that Illumina's lower F1 score is mainly due to recall rather than precision (*Figure 2—figure supplement 2*). We hypothesised that Illumina errors are related to alignment difficulties in repetitive or variant-dense regions due to its shorter reads.

*Figure 4* shows that variant density and repetitive regions account for many false negatives, lowering recall. We define variant density as the number of variants (missed or called) in a 100 bp window around each call. *Figure 4a* reveals a bimodal distribution of variant density for Illumina FNs, with a second peak at 20 variants per 100 bp, unlike the distribution for TP and FP calls. In contrast, Clair3, a top-performing ONT variant caller, shows no bimodal distribution and few missed or false calls at this density (*Figure 4b*). Illumina reads struggle to align in variant-dense regions, whereas ONT reads can (*Figure 4—figure supplement 1*), as 20 variants per 100 bp represents a larger portion of an Illumina read than an ONT read.

We also assessed the change in F1 score when masking repetitive regions of the genome (see Identifying repetitive regions). Due to their shorter length, Illumina reads struggle more with alignment in these regions compared to ONT reads (*Treangen and Salzberg, 2011*). *Figure 4—figure supplement 2* highlights missed variants and alignment gaps in Illumina data. This is further quantified by the increase in Illumina's F1 score when repetitive regions are masked (*Figure 4c*), rising from 99.3% to 99.7%. In contrast, Clair3 100× simplex sup data shows only a 0.003% increase.

In terms of ONT missed calls, a variant-dense repetitive region in the *E. coli* sample ATCC_25922 was the cause of the simplex sup SNP outlier from *Figure 2* (see Appendix 2). In addition, the duplex sup SNP outlier was caused by very low read depth for sample KPC2_202310 (*K. pneumoniae*; Appendix 2).

Indels have traditionally been a systematic weakness for ONT sequencing data, primarily driven by variability in the length of homopolymeric regions as determined by basecallers (*Delahaye and Nicolas, 2021*). Having seen the drastic improvements in read accuracy in *Figure 1*, we sought to determine whether FP indel calls are still a byproduct of homopolymer-driven errors.

When analysing Clair3, the best-performing ONT caller, we found that reads basecalled with the fast model often miscalculate homopolymer lengths by 1 or 2 bp (*Figure 5*), though there is an equal number of non-homopolymeric false indel calls. In contrast, the sup model significantly reduced false indel calls, matching Illumina's error profile. Of the eight false indel calls by Clair3 on sup data, five were homopolymers and three occurred within one or two bases of another insertion with a similar

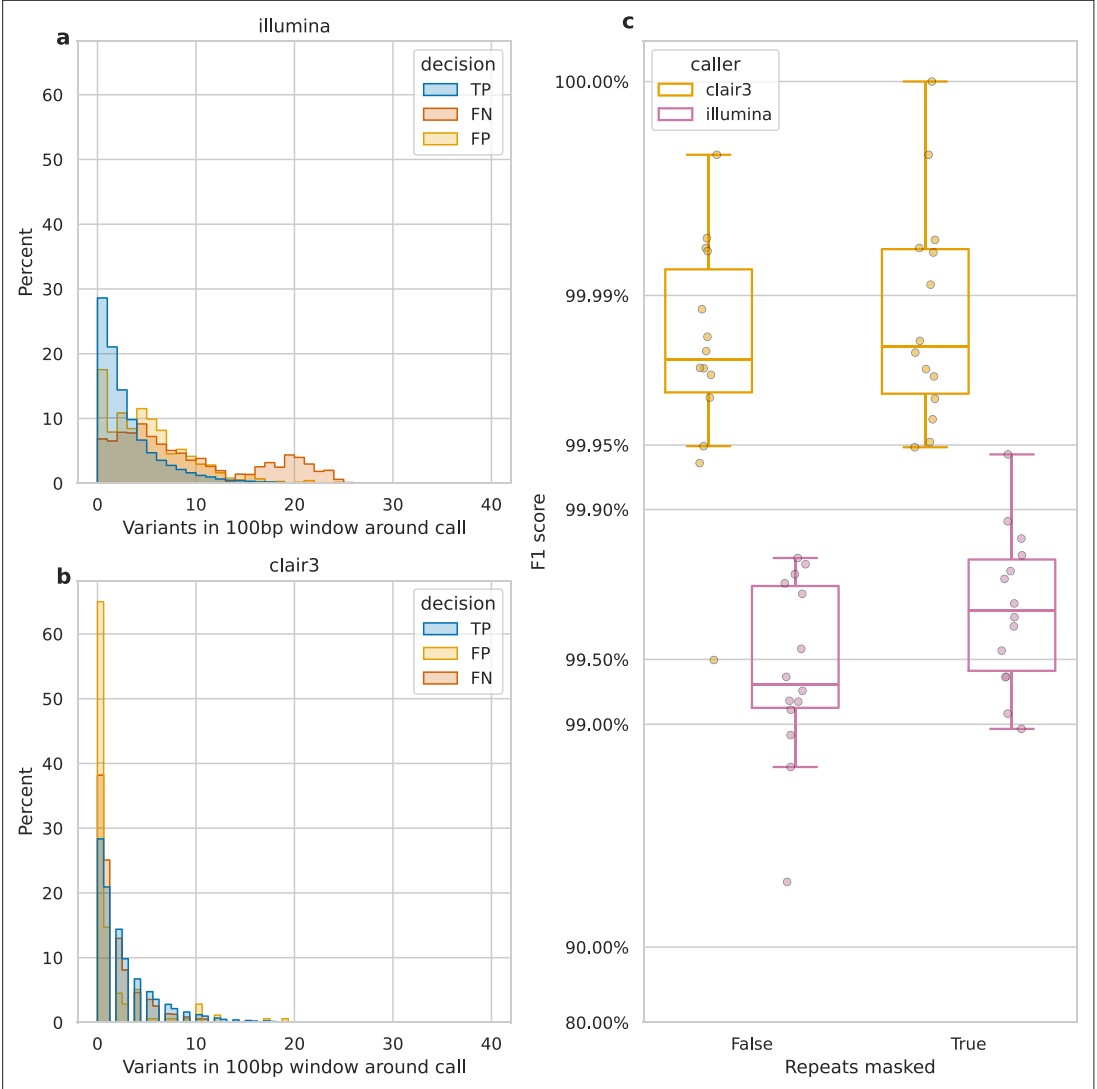

**Figure 4.** Impact of variant density and repetitive regions on Illumina variant calling. Variant density is the number of (true or false) variants in a 100 bp window centred on a call. (**a** and **b**) The distribution of variant densities for true positive (TP), false positive (FP), and false negative (FN) calls. The y-axis, percent, indicates the percent of all calls of that decision that fall within the density bin on the x-axis. Illumina calls, aggregated across all samples, are shown in a, while b shows Clair3 calls from simplex sup-basecalled reads at 100× depth. (**c**) Impact of repetitive regions on the F1 score (y-axis) for Clair3 (100× simplex sup) and Illumina. The x-axis indicates whether variants that fall within repetitive regions are excluded from the calculation of the F1 score. Points indicate the F1 score for a single sample.

The online version of this article includes the following figure supplement(s) for figure 4:

**Figure supplement 1.** Read pileup in a variant-dense region in the genome of sample ATCC_17802__202309.

**Figure supplement 2.** Read pileup around two repetitive regions (horizontal blue bars) in the genome of sample AMtb_1__202401.

sequence. The hac model improved over the fast model but still produced notable false indel calls, mainly miscalculating homopolymers by 1 bp. DeepVariant showed a similar error profile to Clair3 (***Figure 5—figure supplement 2***), with 8/11 false indels being homopolymers. FreeBayes (***Figure 5— figure supplement 3***), Medaka (***Figure 5—figure supplement 4***), and NanoCaller (***Figure 5—figure supplement 5***) performed similarly, while BCFtools (***Figure 5—figure supplement 1***) exhibited a persistent bias for homopolymeric indel errors, even with sup model reads. This indicates that while the sup basecaller reduces bias, deep learning methods like Clair3 and DeepVariant further mitigate it by training models to account for these systematic issues. An honourable mention goes to FreeBayes, a traditional variant caller that handles errors without inherent bias.

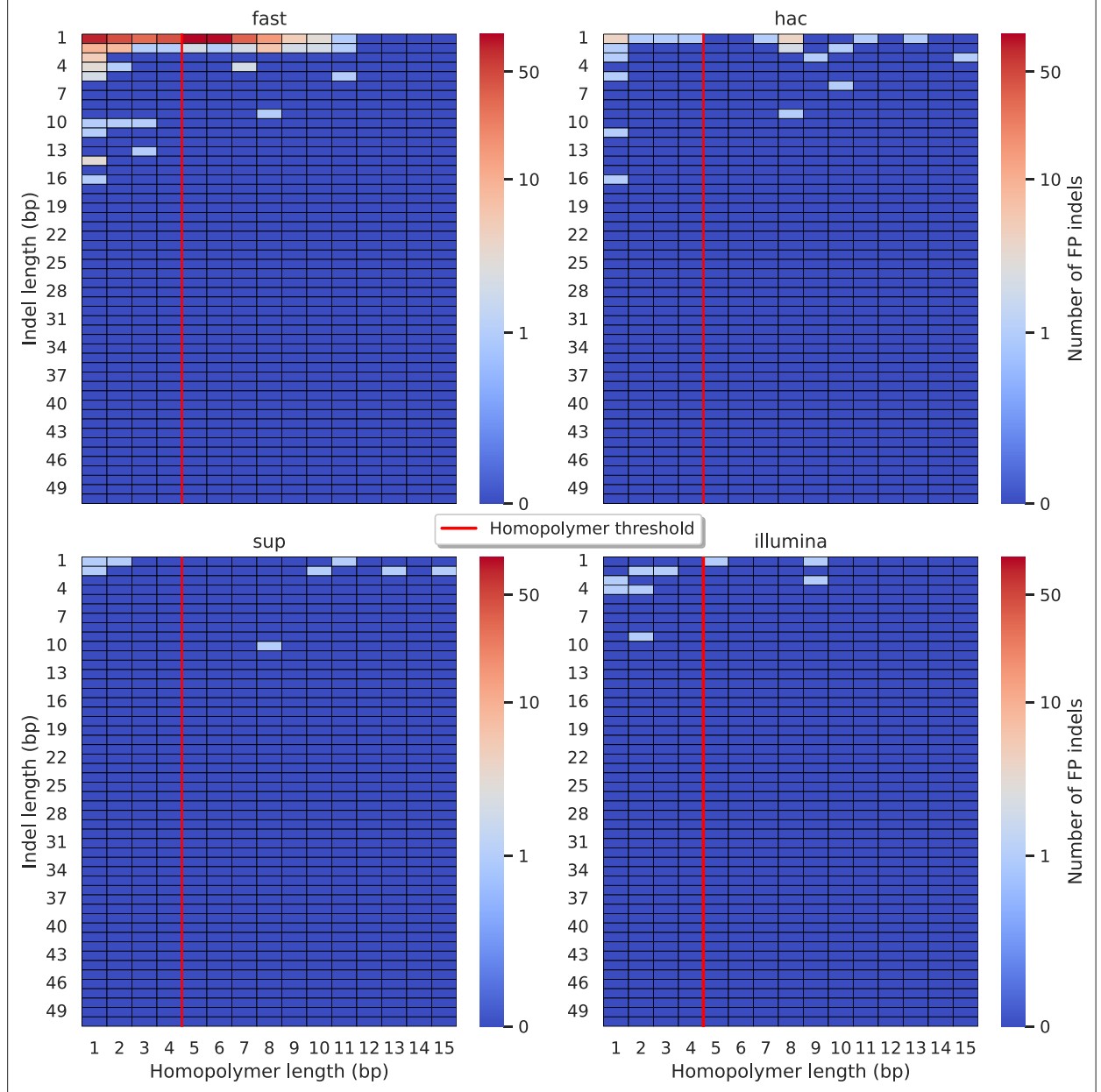

**Figure 5.** Relationship between indel length (y-axis) and homopolymer length (x-axis) for false positive (FP) indel calls for Clair3 100× simplex fast (top left), hac (top right), and sup (lower left) calls. Illumina is shown in the lower right for reference. The vertical red line indicates the threshold above which we deem a run of the same nucleotide to be a 'true' homopolymer. Indel length is the number of bases inserted/deleted for an indel, whereas the homopolymer length indicates how long the tract of the same nucleotide is after the indel. The colour of a cell indicates number of FP indels of that indel-homopolymer length combination.

The online version of this article includes the following figure supplement(s) for figure 5:

**Figure supplement 1.** Relationship between indel length (x-axis) and homopolymer length (y-axis) for false positive (FP) indel calls for BCFtools 100× simplex fast (top left), hac (top right), and sup (lower left) calls.

**Figure supplement 2.** Relationship between indel length (x-axis) and homopolymer length (y-axis) for false positive (FP) indel calls for DeepVariant 100× simplex fast (top left), hac (top right), and sup (lower left) calls.

**Figure supplement 3.** Relationship between indel length (x-axis) and homopolymer length (y-axis) for false positive (FP) indel calls for FreeBayes 100× simplex fast (top left), hac (top right), and sup (lower left) calls.

**Figure supplement 4.** Relationship between indel length (x-axis) and homopolymer length (y-axis) for false positive (FP) indel calls for Medaka 100× simplex fast (top left), hac (top right), and sup (lower left) calls.

*Figure 5 continued on next page*

*Figure 5 continued*

**Figure supplement 5.** Relationship between indel length (x-axis) and homopolymer length (y-axis) for false positive (FP) indel calls for NanoCaller 100× simplex fast (top left), hac (top right), and sup (lower left) calls.

**Figure supplement 6.** Relationship between indel length (x-axis) and homopolymer length (y-axis) for false negative (FN) indel calls for Clair3 100× simplex fast (top left), hac (top right), and sup (lower left) calls.

**Figure supplement 7.** Relationship between indel length (x-axis) and homopolymer length (y-axis) for false negative (FN) indel calls for BCFtools 100× simplex fast (top left), hac (top right), and sup (lower left) calls.

**Figure supplement 8.** Relationship between indel length (x-axis) and homopolymer length (y-axis) for false negative (FN) indel calls for DeepVariant 100× simplex fast (top left), hac (top right), and sup (lower left) calls.

**Figure supplement 9.** Relationship between indel length (x-axis) and homopolymer length (y-axis) for false negative (FN) indel calls for FreeBayes 100× simplex fast (top left), hac (top right), and sup (lower left) calls.

**Figure supplement 10.** Relationship between indel length (x-axis) and homopolymer length (y-axis) for false negative (FN) indel calls for Medaka 100× simplex fast (top left), hac (top right), and sup (lower left) calls.

**Figure supplement 11.** Relationship between indel length (x-axis) and homopolymer length (y-axis) for false negative (FN) indel calls for NanoCaller 100× simplex fast (top left), hac (top right), and sup (lower left) calls.

Lastly, we did not see any systematic indel bias in the context of missed calls (*Figure 5—figure supplements 6–11*), especially when compared to Illumina indel error profiles.

## How much read depth is enough?

Having established the accuracy of variant calls from 'full-depth' ONT datasets (100×), we investigated the required ONT read depth to achieve desired precision or recall, which varies by use case and resource availability. This is particularly relevant for ONT, where sequencing can be stopped in real time once 'sufficient' data is obtained.

We subsampled each ONT read set with rasusa (v0.8.0, *Hall, 2022*) to average depths of 5×, 10×, 25×, 50×, and 100× and called variants with these reduced sets. Due to limited duplex depth, 50× was the maximum used for duplex reads, while 100× was used for simplex reads.

*Figure 6* and *Figure 7* show F1 score, precision, and recall as functions of read depth for SNPs and indels. Precision and recall decrease as read depth is reduced, notably below 25×. Remarkably, Clair3 or DeepVariant on 10× ONT sup simplex data provides F1 scores consistent with, or better than, full-depth Illumina for both SNPs and indels (see *Supplementary file 1a* for Illumina read depths). The same is true for duplex hac or sup reads.

With 5× of ONT read depth the F1 score is lower than Illumina for almost all variant caller and basecalling models. However, BCFtools surprisingly produces SNP F1 scores on par with Illumina on duplex sup reads. Despite the inferior F1 scores across the board at 5×, SNP precision remains above Illumina with duplex reads for all methods except NanoCaller, and calls from Clair3 and DeepVariant simplex sup data.

## What computational resources do I need?

The final consideration for variant calling is the required computational resources. While this may be trivial for those with high-performance computing (HPC) access, many analyse bacterial genomes on personal computers due to their smaller size compared to eukaryotes. The main resource constraints are memory and runtime, especially for aligning reads to a reference and calling variants. Additionally, if working with raw (pod5) ONT data, basecalling is also a resource-intensive step.

*Figure 8* shows the runtime (seconds per megabase of sequencing data) and maximum memory usage for read alignment and variant calling (see *Figure 8—figure supplement 1* and *Supplementary file 1g* for basecalling GPU runtimes). DeepVariant was the slowest (median 5.7 s/Mbp) and most memory-intensive (median 8 GB), with a runtime of 38 min for a 4 Mbp genome at 100× depth. FreeBayes had the largest runtime variation, with a maximum of 597 s/Mbp, equating to 2.75 days for the same genome. In contrast, basecalling with a single GPU using the super-accuracy model required a median runtime of 0.77 s/Mbp, or just over 5 min for a 4 Mbp genome at 100× depth. Clair3 had a median memory usage of 1.6 GB and a runtime of 0.86 s/Mbp (<6 min for a 4 Mbp 100× genome). Full details are given in *Supplementary file 1f*.

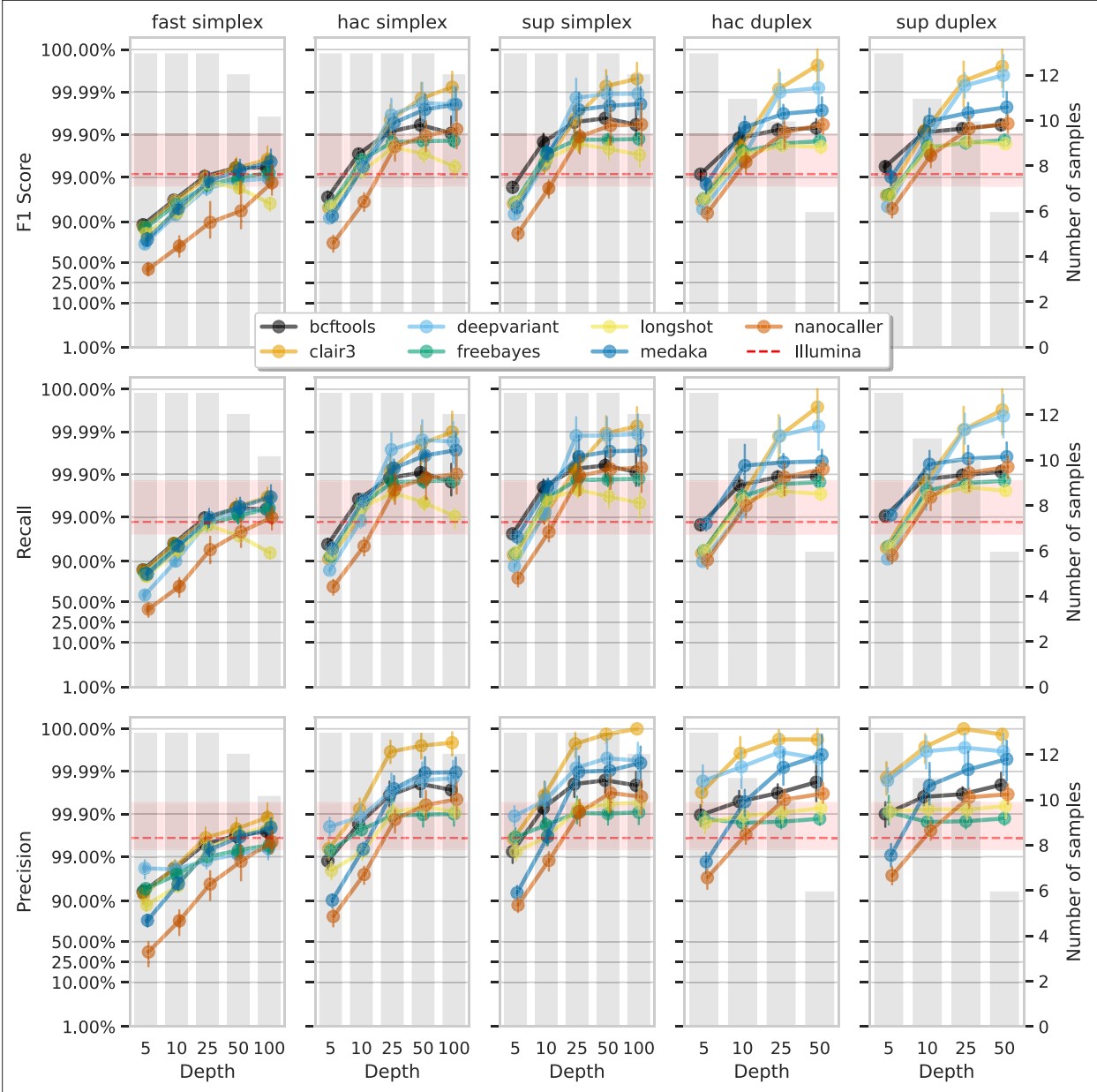

**Figure 6.** Effect of read depth (x-axis) on the highest SNP F1 score, and precision and recall at that F1 score (y-axis), for each variant caller (colours). Each column is a basecall model and read type combination. The grey bars indicate the number of samples with at least that much read depth in the full read set. Samples with less than that depth were not used to calculate that depth's metrics. Bars on each point at each depth depict the 95% confidence interval. The horizontal red dashed line is the full-depth Illumina value for that metric, with the red bands indicating the 95% confidence interval.

## Discussion

In this study, we evaluated the accuracy of bacterial variant calls derived from ONT using both conventional and deep learning-based tools. Our findings show that deep learning approaches, specifically Clair3 and DeepVariant, deliver high accuracy in SNP and indel calls from the latest high-accuracy basecalled ONT data, outperforming Illumina-based methods, with Clair3 achieving median F1 scores of 99.99% for SNPs and 99.53% for indels.

Our dataset comprised deep sequencing of 14 bacterial species using the latest ONT R10.4.1 flow cells, with a 5 kHz sampling rate and complementary deep Illumina sequencing. Consistent with previous studies (*Sanderson et al., 2024*; *Sanderson et al., 2023*; *Sereika et al., 2022*), we observed

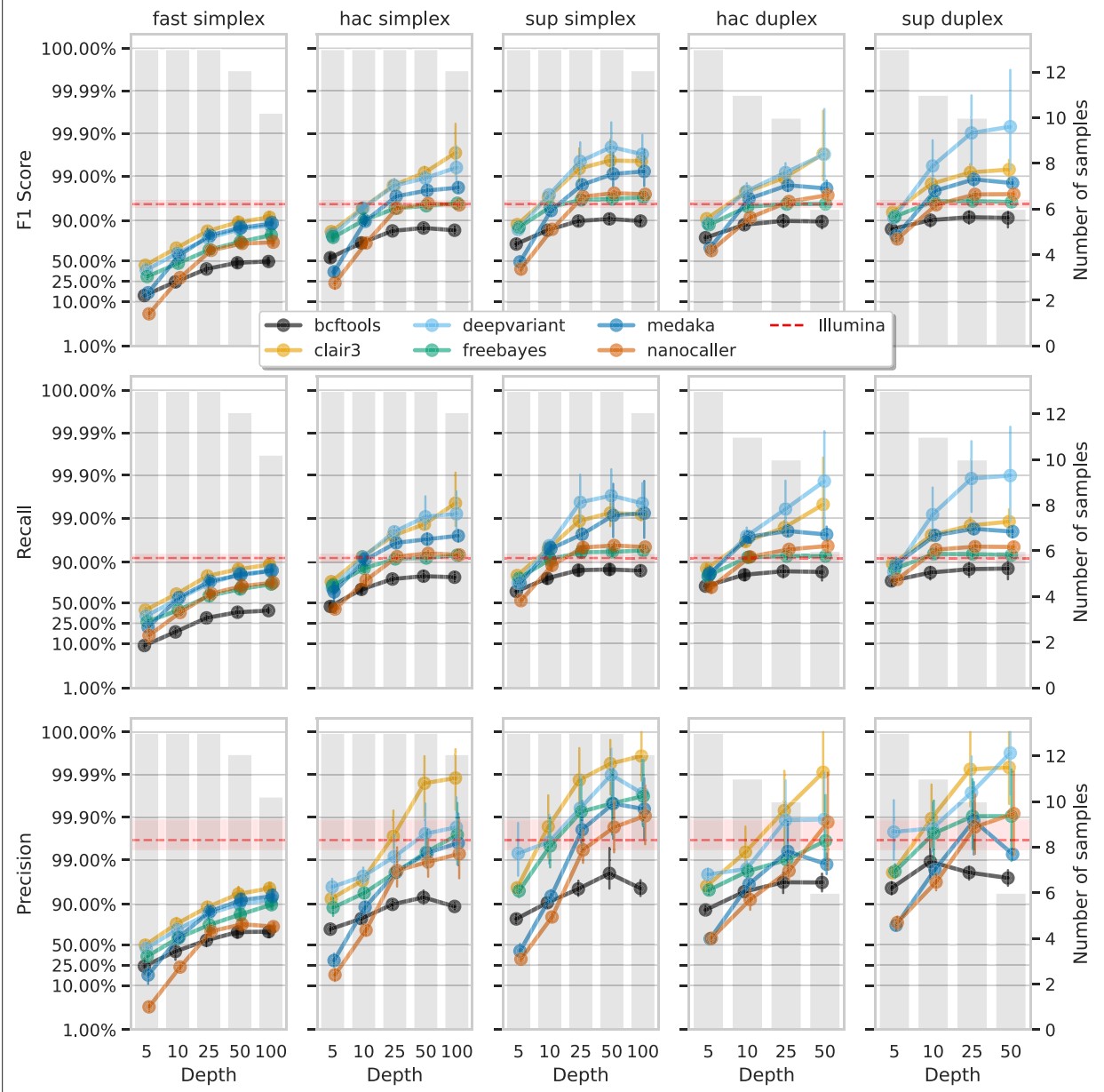

**Figure 7.** Effect of read depth (x-axis) on the highest indel F1 score, and precision and recall at that F1 score (y-axis), for each variant caller (colours). Each column is a basecall model and read type combination. The grey bars indicate the number of samples with at least that much read depth in the full read set. Samples with less than that depth were not used to calculate that depth's metrics. Bars on each point at each depth depict the 95% confidence interval. The horizontal red dashed line is the full-depth Illumina value for that metric, with the red bands indicating the 95% confidence interval.

read accuracies greater than 99.0% (Q20) and 99.9% (Q30) for simplex and duplex reads, respectively (*Figure 1*).

The high-quality sequencing data enabled the creation of near-perfect reference genomes, crucial for evaluating variant calling accuracy. While not claiming perfection for these genomes, we consider them to be as accurate as current technology allows (or as philosophically possible) (*Wick et al., 2023*; *Sereika et al., 2022*).

To benchmark variant calling, we utilised a variant truthset generated by applying known differences between closely related genomes to a reference. This pseudo-real method offers a realistic evaluation framework and a reliable truthset for assessing variant calling accuracy (*Li, 2014*; *Li et al., 2018*).

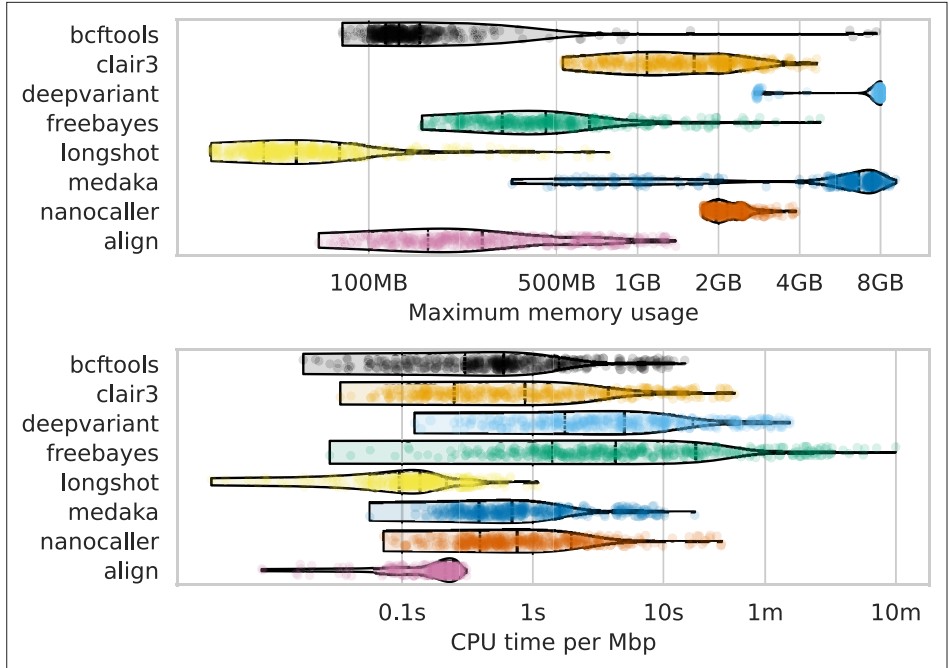

**Figure 8.** Computational resource usage of alignment and each variant caller (y-axis and colours). The top panel shows the maximum memory usage (x-axis) and the lower panel shows the runtime as a function of the CPU time (seconds) divided by the number of basepairs in the readset (seconds per megabasepairs; x-axis). Each point represents a single run across read depths, basecalling models, read types, and samples for that variant caller (or alignment). s=seconds; m=minutes; MB = megabytes; GB = gigabytes; Mbp = megabasepairs.

The online version of this article includes the following figure supplement(s) for figure 8:

**Figure supplement 1.** Runtime of basecalling Oxford Nanopore Technologies (ONT) data with different models on GPUs.

Our comparison of variant calling methods showed that deep learning techniques achieved the highest F1 scores for SNP and indel detection, indicating their potential in genomic analyses and suggesting a shift towards more advanced computational approaches. While the superior performance of these methods has been established for human variant calls (*Olson et al., 2022*; *Olson et al., 2023*), our results confirm their effectiveness for bacterial genomes as well.

Our investigation into missed and false variant calls highlights inherent challenges posed by sequencing technology limitations, particularly read length, alignment in complex regions, and indel length in homopolymers. We found that variant density and repetitive regions hinder Illumina variant calling due to short read alignment issues. However, we found recent improvements in ONT read accuracy and deep learning-based variant callers have mitigated homopolymer-induced FP indel calls, previously a major systematic issue with ONT data (*Delahaye and Nicolas, 2021*; *Sereika et al., 2022*).

Having established the accuracy and error sources of modern methods, we examined the impact of read depth on variant calling accuracy. Our results show that high accuracy is achievable at reduced read depths of 10×, especially with super-accuracy basecalling models and deep learning algorithms. This is significant for resource-limited projects, as 10× super-accuracy simplex data can match or exceed Illumina accuracy. For optimal clinical and public health applications, we recommend a minimum of 25× depth. Notably, 5× depth with duplex super-accuracy ONT data achieved SNP accuracy comparable to Illumina. Having such confidence in low-depth calls will no doubt be a boon for many clinical and public health applications where sequencing direct-from-sample is desired (*Sheka et al., 2021*; *Street et al., 2020*; *Chiu and Miller, 2019*; *Nilgiriwala et al., 2023*).

Lastly, considering computational resource requirements is crucial, especially for those without HPC facilities (*Musila, 2022*; *Hoenen et al., 2016*; *Faria et al., 2016*). Our findings show a wide range of demands among variant calling methods, with the worst-case scenario (FreeBayes) taking over 2

days. Most methods, however, run in less than 40 min, with Clair3 having a median runtime of about 6 minutes. All methods use less than 8 GB of memory, making them compatible with most laptops. Basecalling is generally faster than variant calling, assuming GPU access, which is likely considered when acquiring ONT-related equipment.

There are three main limitations to this work. The first is that we only assess small variants and ignored structural variants. Zhou et al. benchmarked structural variant calling from ONT data (*Zhou et al., 2019*), though this focused on human sequencing data. Generating a truthset of structural variants between two genomes is, in itself, a difficult task. However, we believe such an undertaking with a thorough investigation of structural variant calling methods for bacterial genomes would be highly beneficial.

The second limitation is not using a diverse range of ANI values for selecting the variant donor genomes when generating the truthset. Previous work from Bush et al. examined different diversity thresholds for selecting reference genomes when calling variants from Illumina data, and found it to be one of the main differentiating factors in accuracy (*Bush et al., 2020*). Our results mirror this to an extent, showing the reduction in Illumina accuracy as the variant density increases, though it would be interesting to determine whether the divergence in reference genomes has an affect on ONT variant calling accuracy. Nevertheless, to maintain our focus on the nuances of variant calling methods, including basecalling models, read types, error types, and the influence of read depth, we decided that introducing another layer of complexity into our benchmark could potentially obscure some of the insights.

The third limitation is that Illumina sequencing was performed on different models: three samples on the NextSeq 500 and the rest on the NextSeq 2000. While differences in error rates exist between Illumina instruments, no specific assessment has been made between these NextSeq models (*Stoler and Nekrutenko, 2021*). However, the absolute differences in error rates are minor and unlikely to impact our study significantly. This is particularly relevant since Illumina's lower F1 score compared to ONT was due to missed calls rather than erroneous ones.

In conclusion, this study comprehensively evaluates bacterial variant calls using ONT, highlighting the superior performance of deep learning tools, particularly Clair3 and DeepVariant, in SNP and indel detection. Our extensive dataset and rigorous benchmarking demonstrate significant advancements in sequencing accuracy with the latest ONT technologies. Improvements in ONT read accuracy and deep learning variant callers have mitigated previous challenges like homopolymer-associated errors. We also found that high accuracy can be achieved at lower read depths, making these methods practical for resource-limited settings. This capability marks a significant step in making advanced genomic analysis more accessible and impactful.

## Methods
### Sequencing

Bacterial isolates were streaked onto agar plates and grown overnight at 37°C. *M. tuberculosis*, *S. pyogenes*, and *S. dysgalactiae* subsp. *equisimilis* were grown in liquid media of 7H9 or TSB with shaking until reaching high cell density (OD ~ 1; see Appendix 1 for *Streptococcus* sample selection). The cultures were centrifuged at 13,000 rpm for 10 min and cell pellets were collected. Bacteria were lysed with appropriate enzymatic treatment except for *Mycobacterium* and *Streptococcus*, which were lysed by bead beating (PowerBead, 0.5 mm glass beads [13116-50] or Lysing Matrix Y [116960050-CF] and Precellys or TissueLyser [QIAGEN]). DNA extraction was performed by sodium acetate precipitation and further Ampure XP bead purification (Beckman Coulter) or either Beckman Coulter GenFind V2 (A41497) or QIAGEN Blood and Tissue DNEasy kit (69506). Illumina library preparation was performed using Illumina DNA prep (20060059) using quarter reagents and Illumina DNA/RNA UD Indexes. Short-read whole-genome sequencing was performed on an Illumina NextSeq 500 for the *M. tuberculosis* (AMtb_1__202402), *S. pyogenes* (RDH275__202311), and *S. dysgalactiae* (MMC234__202311) samples and a NextSeq 2000 for all other samples, with a 150 bp PE kit. ONT library preparation was performed using either Rapid Barcoding Kit V14 (SQK-RBK114.96) or Native Barcoding Kit V14 (SQK-NBD114.96). Long-read whole-genome sequencing was performed on a MinION Mk1b or GridION using R10.4.1 MinION flow cells (FLO-MIN114). *Supplementary file 1i* contains detailed information about the instrument models and multiplexing for each sample.

## Basecalling and quality control

Raw ONT data were basecalled with Dorado (v0.5.0, *Oxford Nanopore Technologies, 2023a*) using the v4.3.0 models *fast* (dna_r10.4.1_e8.2_400bps_fast@v4.3.0), *hac* (dna_r10.4.1_e8.2_400bps_hac@v4.3.0), and *sup* (dna_r10.4.1_e8.2_400bps_sup@v4.3.0). Duplex reads were additionally generated using the `duplex` subcommand of Dorado with hac and sup models (fast is not compatible with duplex). All runs were basecalled on an Nvidia A100 GPU to ensure consistency. Reads shorter than 1000bp or with a quality score below 10 were removed with SeqKit (v2.6.1 *Shen et al., 2016*) and each readset was randomly subsampled to a maximum mean read depth of 100x with Rasusa (v0.8.0 *Hall, 2022*).

Illumina reads were preprocessed with fastp (v0.23.4, *Chen et al., 2018*) to remove adapter sequences, trim low-quality bases from the ends of the reads, and remove duplicate reads and reads shorter than 30bp.

## Genome assembly

Ground truth assemblies were generated for each sample as per *Wick et al., 2023*. Briefly, the unfiltered ONT simplex sup reads were filtered with Filtlong (v0.2.1, *Wick, 2021*) to keep the best 90% (`-p 90`) and fastp (default settings) was used to process the raw Illumina reads. We performed 24 separate assemblies using the Extra-thorough assembly instructions in Trycycler's (v0.5.4, *Wick et al., 2021*) documentation. Assemblies were combined into a single consensus assembly with Trycycler and Illumina reads were used to polish that assembly using Polypolish (v0.6.0; default settings, *Wick and Holt, 2022*) and Pypolca (v0.3.1, *Bouras et al., 2024*; *Zimin and Salzberg, 2020*) with `--careful`. Manual curation and investigation of all polishing changes was made as per *Wick et al., 2023* (e.g. for very long homopolymers, the correct length was chosen as per Illumina reads support).

## Truthset and reference generation

To generate the variant truthset for each sample, we identified all variants between the sample and a *donor* genome. To select the variant-donor genome for a given sample, we downloaded all RefSeq assemblies for that species (up to a maximum of 10,000) using genome_updater (v0.6.3, *Piro, 2023*). ANI was calculated between each downloaded genome and the sample reference using skani (v0.2.1, *Shaw and Yu, 2023*). We only kept genomes with an ANI, $a$, such that $98.40\% \leq a <= 99.80\%$. In addition, we excluded any genomes with CheckM (*Parks et al., 2015*) completeness less than 98% and contamination greater than 5%. We then selected the genome with the ANI closest to 99.50%. Our reasoning for this range exclusion is that genomes with $a > 99.80\%$ are almost always members of the same sequence type (ST) (*Rodriguez-R et al., 2024*; *Viver et al., 2024*), and we found very little variation between them (data not shown).

We then identified variants between the reference and donor genomes using both minimap2 (v2.26, *Li, 2018*) and mummer (v4.0.0rc1, *Marçais et al., 2018*). We took the intersection of the variants identified by minimap2 and mummer into a single variant call file (VCF) and used BCFtools (v1.19, *Danecek et al., 2021*) to decompose multi-nucleotide polymorphisms (MNPs) into SNPs, left-align and normalise indels, remove duplicate and overlapping variants, and exclude any indel longer than 50 bp. The resulting VCF file is our truthset.

Next, we generated a mutated reference genome, which we used as the reference against which variants were called by the different methods we assess. BCFtools' `consensus` subcommand was used to apply the truthset of variants to the sample reference, thus producing a mutated reference.

## Alignment and variant calling

ONT reads were aligned to the mutated reference with minimap2 using options `--cs --MD -aLx map-ont` and output to a BAM alignment file.

Variant calling was performed from the alignment files with BCFtools (v1.19, *Danecek et al., 2021*), Clair3 (v1.0.5, *Zheng et al., 2022*), DeepVariant (v1.6.0, *Poplin et al., 2018*), FreeBayes (v1.3.7, *Garrison, 2012*), Longshot (v0.4.5, *Edge and Bansal, 2019*), and NanoCaller (v3.4.1, *Ahsan et al., 2021*). In addition, variant calling was performed directly from the reads for Medaka (v1.11.3, *Oxford Nanopore Technologies, 2023b*) as Medaka does its own alignment with minimap2. Individual parameters used for each variant caller can be found in the accompanying GitHub repository (*Hall, 2024a*; *Hall, 2023*).

Where a variant caller provided an option to set the expected ploidy, haploid was given. In addition, where a minimum read depth or base quality option was available, a value of 2 and 10, respectively, was used in order to try and make downstream assessment and filtering consistent across callers.

For Clair3, the pretrained models for Dorado model v4.3.0 provided by ONT were used (*Oxford Nanopore Technologies, 2023c*). However, as no fast model is available, we used the hac model with the fast-basecalled reads. The pretrained model option `--model_type ONT_R104` was used with DeepVariant, and the default model was used for NanoCaller. For Medaka, the provided v4.3.0 sup and hac models were used, with the hac model being used for fast data as no fast model is available.

For the Illumina variant calls that act as a benchmark to compare ONT against, we chose Snippy (*Seemann, 2015*) due to it being tailored for haploid genomes and being one of the best performing variant callers on Illumina data (*Bush et al., 2020*). Snippy performs alignment of reads with BWA-MEM (*Li, 2013*) and calls variants with FreeBayes.

VCFs were then filtered to remove overlapping variants, make heterozygous calls homozygous for the allele with the most depth, normalise and left-align indels, break MNPs into SNPs, and remove indels longer than 50 bp, all with BCFtools.

## Variant call assessment

Filtered VCFs were assessed with vcfdist (v2.3.3, *Dunn and Narayanasamy, 2023*) using the truth VCFs and mutated references from Truthset and reference generation. We disabled partial credit with `--credit-threshold 1.0` and set the maximum variant quality threshold (`-mx`) to the maximum in the VCF being assessed.

## Identifying repetitive regions

To identify repetitive regions in the mutated reference, we used the following mummer utilities. `nucmer --maxmatch --nosimplify` to align the reference against itself and retain non-unique alignments. We then passed the output into `show-coords -rTH -I 60` to obtain the coordinates for all alignments with an identity of 60% or greater. Alignments where the start and end coordinates of the alignment do not match are considered as repeats and these are output in the BED format, with intervals being merged with BEDtools (*Quinlan and Hall, 2010*).

## Code availability

All code to perform the analyses in this work are available on GitHub and archived on Zenodo (*Hall, 2024a*; *Hall, 2023*).

## Acknowledgements

This research was performed in part at Doherty Applied Microbial Genomics, Department of Microbiology and Immunology, The University of Melbourne at the Peter Doherty Institute for Infection and Immunity. This research was supported by the University of Melbourne's Research Computing Services and the Petascale Campus Initiative. We are grateful to Bart Currie and Tony Korman for providing the *S. dysgalactiae* (MMC234__202311) and *S. pyogenes* (RDH275__202311) samples. We thank Zamin Iqbal and Martin Hunt for insightful discussions relating to truthset and reference generation. We would also like to thank Romain Guérillot and Miranda Pitt for invaluable feedback throughout the project. Funding: This work was supported by the Australian Government Medical Research Future Fund (MRFF) Genomics Health Futures Mission (GHFM) Flagships – Pathogen Genomics Grant (FSPGN000045) META-GP: DELIVERING A CLINICAL METAGENOMICS PLATFORM FOR AUSTRALIA. The funding body had no role in the design, analysis, interpretation, or writing of this work.

## Additional information

### Competing interests

Lachlan Coin: Has received support from ONT to present his findings at scientific conferences; ONT played no role in study design, execution, analysis, or publication. Received research funding from ONT unrelated to this project. The other authors declare that no competing interests exist.

## Funding

| Funder | Grant reference number | Author |
| --- | --- | --- |
| National Health and Medical Research Council | FSPGN000045 | Michael B Hall Eike J Steinig |

The funders had no role in study design, data collection and interpretation, or the decision to submit the work for publication.

## Author contributions

Michael B Hall, Conceptualization, Data curation, Software, Formal analysis, Investigation, Visualization, Methodology, Writing – original draft, Project administration, Writing – review and editing; Ryan R Wick, Conceptualization, Data curation, Software, Formal analysis, Investigation, Methodology, Writing – original draft, Writing – review and editing; Louise M Judd, Conceptualization, Resources, Data curation, Investigation, Methodology, Writing – original draft, Project administration, Writing – review and editing; An N Nguyen, Resources, Investigation, Methodology, Writing – original draft, Writing – review and editing; Eike J Steinig, Conceptualization, Data curation, Software, Formal analysis, Investigation, Methodology, Writing – review and editing; Ouli Xie, Resources, Methodology, Writing – review and editing; Mark Davies, Resources, Supervision, Methodology, Writing – review and editing; Torsten Seemann, Conceptualization, Supervision, Methodology, Writing – review and editing; Timothy P Stinear, Conceptualization, Resources, Supervision, Funding acquisition, Methodology, Writing – review and editing; Lachlan Coin, Conceptualization, Resources, Supervision, Funding acquisition, Methodology, Writing – original draft, Writing – review and editing

## Author ORCIDs

Michael B Hall ⓘ https://orcid.org/0000-0003-3683-6208
Ryan R Wick ⓘ http://orcid.org/0000-0001-8349-0778
Louise M Judd ⓘ https://orcid.org/0000-0003-3613-4839
An N Nguyen ⓘ https://orcid.org/0000-0003-4893-6636
Ouli Xie ⓘ http://orcid.org/0000-0002-5032-1932
Timothy P Stinear ⓘ https://orcid.org/0000-0003-0150-123X
Lachlan Coin ⓘ https://orcid.org/0000-0002-4300-455X

Reviewer #2 (Public review): https://doi.org/10.7554/eLife.98300.3.sa1
Reviewer #3 (Public review): https://doi.org/10.7554/eLife.98300.3.sa2
Author response https://doi.org/10.7554/eLife.98300.3.sa3

# Additional files

## Supplementary files

• Supplementary file 1. Supplementary tables. (**a**) Sequencing read quality metrics. (**b**) Full details of the donor genomes summarised in *Table 1*. (**c**) Summary of the F1 score, precision, and recall at the best F1 score for each variant caller. (**d**) Variant calling results for each sample across all conditions. (**e**) Variant call file (VCF) quality scores that produce the optimal F1 score. (**f**) Summary statistics of CPU and memory usage presented in *Figure 8*. (**g**) GPU basecalling runtime statistics. (**h**) Accession numbers and DOIs for each sample's sequencing data. (**i**) Information about the sequencing strategy for each sample on each platform.

• MDAR checklist

## Data availability

The unfiltered FASTQ, and assembly files generated in this study have been submitted to the NCBI BioProject database under accession numbers PRJNA1087001 and PRJNA1042815. See *Supplementary file 1h* for a list of all Assembly, BioSample and Run accessions, as well as DOIs for raw ONT pod5 data. Variant truthsets and associated data are archived on Zenodo (*Hall, 2024b*).

The following datasets were generated:

| Author(s) | Year | Dataset title | Dataset URL | Database and Identifier |
|---|---|---|---|---|
| Hall MB, Wick RR, Judd LM, Nguyen ANT, Steinig EJ, Xie O, Davies MR, Seemann T, Stinear TP, Coin LJM | 2024 | Sequencing data from: Benchmarking reveals superiority of deep learning variant callers on bacterial nanopore sequence data | https://www.ncbi.nlm.nih.gov/bioproject/PRJNA1087001/ | NCBI BioProject, PRJNA1087001 |
| Wick RR, Judd LM, Stinear TP | 2023 | ATCC genome sequencing | https://www.ncbi.nlm.nih.gov/bioproject/PRJNA1042815/ | NCBI BioProject, PRJNA1042815 |

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

## Appendix 1

### *Streptococcus* sample selection

MMC234__202311 and RDH275__202311 were collected as part of an invasive streptococcal surveillance study in Australia (approved by the Royal Melbourne Hospital Human Research Ethics Committee [HREC/80105/MH-2021] and the Human Research Ethics Committee of the Northern Territory Depart of Health and Menzies School of Health Research [2021-4181]).

## Appendix 2

### Simplex F1 outlier

From *Figure 2* in the main text it is clear that in the simplex SNP panel (top left) there is a single outlier for the sup model for most variant callers. This sample is the same across the variant callers – ATCC_25922 (*E. coli*). We chose to investigate the reason for this outlier using the Clair3 (simplex sup) variant calls. The reason for the lower F1 score was reduced recall (as seen in *Figure 2—figure supplement 2*). ATCC_25922 had 47 FN variants, with 45 of those coming from two repetitive regions (see Identifying repetitive regions in the main text). In addition, these two regions have a higher variant density compared to the rest of the genome and are the repeats of each other. The two regions are 7.5 Kbp long and have an identity of 96.5%. As can be seen from *Figure 1*, there is a lot of heterogeneity at the FN site. This is caused by reads from the reciprocal region multi-mapping to this region. *Appendix 2—figure 2* shows a textual representation of the of the two sequences. Position 3 is the FN, where we expect a C>T variant call. While 50% of the bases at position 3 are indeed T, and 34% are C, Clair3 and most other callers fail to call this site a variant site. This pattern of heterogeneity leading to missed calls was repeated for the other FNs within these repeats. Highlighting that even though deep learning callers using R10 ONT data deal much better with repetitive and variant-dense regions, they can still struggle when you have both of these difficulties combined to a high degree. However, it is interesting to note that Medaka did not have the same issues with missed variants for these sites. Assumably the model training for Medaka is sufficiently different to DeepVariant and Clair3 to allow it to deal with these difficult sites without issue.

The other notable outlier from *Figure 2* in the main text is in the duplex SNP panel (top right). This outlier is sample KPC2_202310 (*K. pneumoniae*). The reason for its reduced F1 score is again to do with recall. However, this reduced recall is simply due to the fact that the duplex depth for this sample is only 3×.

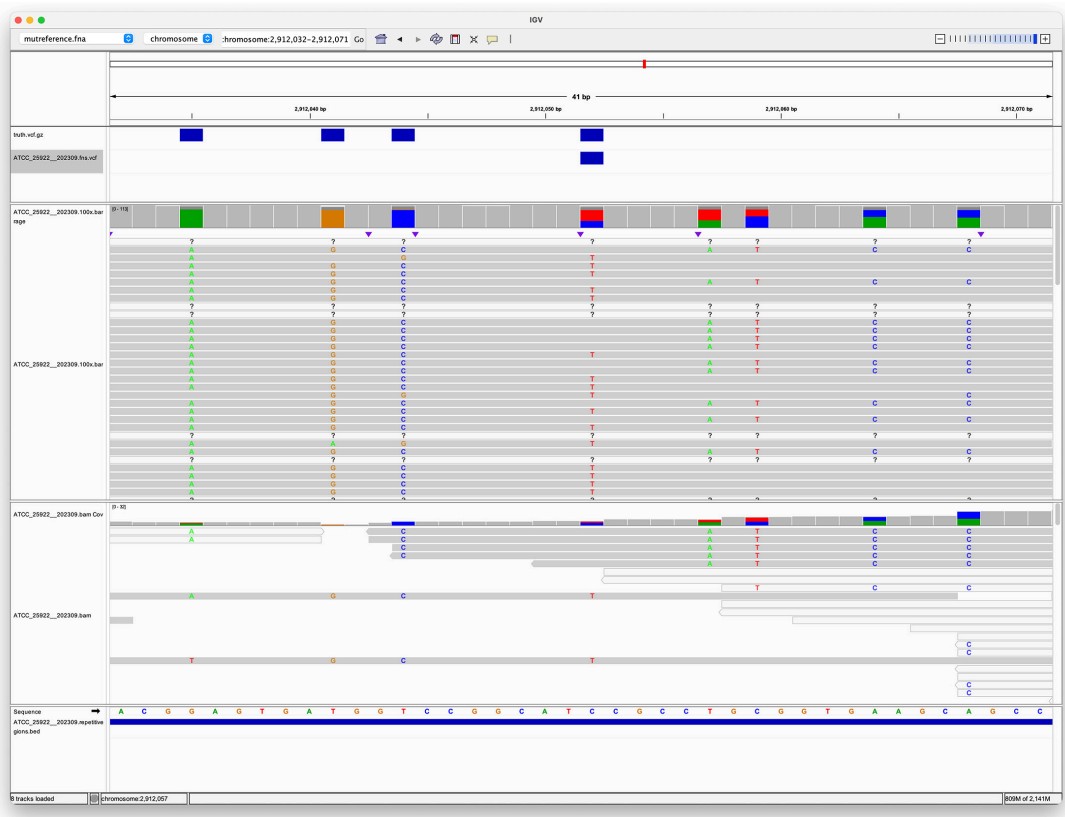

**Appendix 2—figure 1.** Pileup surrounding a Clair3 false negative variant. The top track (four blue rectangles) shows true variants – i.e., variants we expect to find. The second track shows a single false negative variant. The third track shows the alignment of Oxford Nanopore Technologies (ONT) reads to this region, with coloured letters indicating where the aligned sequence disagrees with the reference. The fourth track shows the alignment

of Illumina reads. The fifth track is the reference sequence. The bottom track shows repetitive regions (the whole region is repetitive as the blue rectangle spans the whole region in view).

```
        1       2           3
S1 - {T}GG{T}CCGGCAT{C}CGCC
S2 - {G}GG{C}CCGGCAT{T}CGCC
        1     2           3
```

**Appendix 2—figure 2.** Textual example of two small repetitive regions in sample ATCC_25922 that lead to a false negative at position 3 in sequence S1. Positions 1 and 2 are two other variant positions where a true positive was obtained for S1 (the expected variants at positions 1 and 2 match the sequence in S2).

