## [Editor Report · eLife assessment]

This **important** study shows how a combination of the latest generation of Oxford Nanopore Technology long reads with state-of-the art variant callers enables bacterial variant discovery at an accuracy that matches or exceeds the current "gold standard" with short reads. The work thus heralds a new era, in which Illumina short-read sequencing no longer rules supreme. While the inclusion of a larger number of reference genomes would have enabled an even more fine-grained analysis, the evidence as it is supports the claims of the authors **convincingly**. The work will be of interest to anyone performing sequencing for outbreak investigations, bacterial epidemiology, or similar studies.

---

## [Referee Report · Reviewer #2 (Public review)]

Summary:

Hall et al describe the superiority of ONT sequencing and deep learning-based variant callers to deliver higher SNP and Indel accuracy compared to previous gold-standard Illumina short-read sequencing. Furthermore, they provide recommendations for read sequencing depth and computational requirements when performing variant calling.

Strengths:

The study describes compelling data showing ONT superiority when using deep learning-based variant callers, such as Clair3, compared to Illumina sequencing. This challenges the paradigm that Illumina sequencing is the gold standard for variant calling in bacterial genomes. The authors provide evidence that homopolymeric regions, a systematic and problematic issue with ONT data, are no longer a concern in ONT sequencing.

Weaknesses:

The study is limited in the number of samples included, even though it covers different species with divergent genome sequences, likely covering major evolutionary changes. The methods section could be more detailed. A structural variation analysis would be an interesting next step.

---

## [Referee Report · Reviewer #3 (Public review)]

Hall et al. benchmarked different variant calling methods on Nanopore reads of bacterial samples and compared the performance of Nanopore to short reads produced with Illumina sequencing. To establish a common ground for comparison, the authors first generated a variant truthset for each sample and then projected this set to the reference sequence of the sample to obtain a mutated reference. Subsequently, Hall et al. called SNPs and small indels using commonly used deep learning and conventional variant callers and compared the precision and accuracy from reads produced with simplex and duplex Nanopore sequencing to Illumina data. The authors did not investigate large structural variation, which is a major limitation of the current manuscript. It will be very interesting to see a follow-up study covering this much more challenging type of variation.

In their comprehensive comparison of SNPs and small indels, the authors observed superior performance of deep learning over conventional variant callers when Nanopore reads were basecalled with the most accurate (but also computationally very expensive) model, even exceeding Illumina in some cases. Not surprisingly, Nanopore underperformed compared to Illumina when basecalled with the fastest (but computationally much less demanding) method with the lowest accuracy. The authors then investigated the surprisingly higher performance of Nanopore data in some cases and identified lower recall with Illumina short read data, particularly from repetitive regions and regions with high variant density, as the driver. Combining the most accurate Nanopore basecalling method with a deep learning variant caller resulted in low error rates in homopolymer regions, similar to Illumina data. This is remarkable, as homopolymer regions are (or, were) traditionally challenging for Nanopore sequencing.

Lastly, Hall et al. provided useful information on the required Nanopore read depth, which is surprisingly low, and the computational resources for variant calling with deep learning callers. With that, the authors established a new state-of-the-art for Nanopore-only variant calling on bacterial sequencing data. Most likely these findings will be transferred to other organisms as well or at least provide a proof-of-concept that can be built upon.

As the authors mention multiple times throughout the manuscript, Nanopore can provide sequencing data in nearly real-time and in remote regions, therefore opening up a ton of new possibilities, for example for infectious disease surveillance. In these scenarios, computational resources can be very limited. The highest-performing variant calling method, as established in this study, requires the computationally very expensive sup and/or duplex nanopore basecalling, while the least computationally demanding basecalling method underperforms. To comprehensively guide users through the computational resources required for basecalling and variant calling, the authors provide runtime benchmarks assuming GPU access.

---

## [Author Response]

The following is the authors’ response to the original reviews.

**Public Reviews:**

**Reviewer #1 (Public Review):**
Summary:The authors assess the accuracy of short variant calling (SNPs and indels) in bacterial genomes using Oxford Nanopore reads generated on R10.4 flow cells from a very similar genome (99.5% ANI), examining the impact of variant caller choice (three traditional variant callers: bcftools, freebayes, and longshot, and three deep learning based variant callers: clair3, deep variant, and nano caller), base calling model (fast, hac and sup) and read depth (using both simplex and duplex reads).Strengths:Given the stated goal (analysis of variant calling for reads drawn from genomes very similar to the reference), the analysis is largely complete and results are compelling. The authors make the code and data used in their analysis available for re-use using current best practices (a computational workflow and data archived in INSDC databases or Zenodo as appropriate).Weaknesses:

While the medaka variant caller is now deprecated for diploid calling, it is still widely used for haploid variant calling and should at least be mentioned (even if the mention is only to explain its exclusion from the analysis).

We have now added Medaka haploid caller to the benchmark. It performs quite well overall (better than the traditional methods), but not as good as Clair3 or DeepVariant.

Appraisal:The experiments the authors engaged in are well structured and the results are convincing. I expect that these results will be incorporated into "best practice" bacterial variant calling workflows in the future.

Thank you for the positive appraisal.

**Reviewer #2 (Public Review):**
Summary:Hall et al describe the superiority of ONT sequencing and deep learning-based variant callers to deliver higher SNP and Indel accuracy compared to previous gold-standard Illumina short-read sequencing. Furthermore, they provide recommendations for read sequencing depth and computational requirements when performing variant calling.Strengths:The study describes compelling data showing ONT superiority when using deep learning-based variant callers, such as Clair3, compared to Illumina sequencing. This challenges the paradigm that Illumina sequencing is the gold standard for variant calling in bacterial genomes. The authors provide evidence that homopolymeric regions, a systematic and problematic issue with ONT data, are no longer a concern in ONT sequencing.Weaknesses:(1) The inclusion of a larger number of reference genomes would have strengthened the study to accommodate larger variability (a limitation mentioned by the authors).

Our strategic selection of 14 genomes—spanning a variety of bacterial genera and species, diverse GC content, and both gram-negative and gram-positive species (including M. tuberculosis, which is neither)—was designed to robustly address potential variability in our results. Moreover, all our genome assemblies underwent rigorous manual inspection as the quality of the true genome sequences is the foundation this research is built upon. Given this, the fundamental conclusions regarding the accuracy of variant calls would likely remain unchanged with the addition of more genomes. However, we do acknowledge that a substantially larger sample size, which is beyond the scope of this study, would enable more fine-grained analysis of species differences in error rates.

(2) In Figure 2, there are clearly one or two samples that perform worse than others in all combinations (are always below the box plots). No information about species-specific variant calls is provided by the authors but one would like to know if those are recurrently associated with one or two species. Species-specific recommendations could also help the scientific community to choose the best sequencing/variant calling approaches.

Thank you for highlighting this observation. The precision, recall, and F1 scores for each sample and condition can be found in Supplementary Table S4.

Upon investigation of the outliers in Figure 2 we discovered three things. First, there was a parameter in Longshot we were using that automatically capped coverage and lead to a number of false negatives, leading to its outlier. This has now been rectified and the figure is updated accordingly. Second, the outlier in the simplex sup SNP panel (top left) was the same *E. coli* sample for most variant callers (though Medaka had no issues). The reasoning for this was a variant-dense repetitive region. We have added an in-depth explanation of this, along with figures illustrating the issue in Supplementary Section S2, with a brief statement in the main text. Third, the outlier in the duplex sup SNP panel (top right) is due to a very low (duplex) depth sample. This has also been added briefly to the main text and fully in Section S2.

We have now included a species-segregated version of Figure 2 (Suppl. Figures S5-7) for Clair3 with the sup model (best performer) for a clearer interpretation of how each species performs.

(3) The authors support that a read depth of 10x is sufficient to achieve variant calls that match or exceed Illumina sequencing. However, the standard here should be the optimal discriminatory power for clinical and public health utility (namely outbreak analysis). In such scenarios, the highest discriminatory power is always desirable and as such an F1 score, Recall and Precision that is as close to 100% as possible should be maintained (which changes the minimum read sequencing depth to at least 25x, which is the inflection point).

We agree that the highest discriminatory power is always desirable for clinical or public health applications. In which case, 25x is probably a better minimum recommendation. However, we are also aware that there are resource-limited settings where parity with Illumina is sufficient. In these cases, 10x depth from ONT would provide enough data.

The manuscript previously emphasised the latter scenario, but we have revised the text (Discussion) to clearly recommend 25x depth as a conservative aim in settings where resources are not a constraint, ensuring the highest possible discriminatory power.

(4) The sequencing of the samples was not performed with the same Illumina and ONT method/equipment, which could have introduced specific equipment/preparation artefacts that were not considered in the study. See for example https://academic.oup.com/nargab/article/3/1/lqab019/6193612.

To our knowledge, there is no evidence that sequencing on different ONT machines or barcoding kits leads to a difference in read characteristics or accuracy. To ensure consistency and minimise potential variability, we used the same ONT flowcells for all samples and performed basecalling on the same Nvidia A100 GPU. We have updated the methods to emphasise this.

For Illumina and ONT, the exact machines and kits used for each sample have been added as supplementary table S9 We have also added a short paragraph about possible Illumina error rate differences in the ‘Limitations’ section of the Discussion.

The third limitation is that Illumina sequencing was performed on different models: three samples on the NextSeq 500 and the rest on the NextSeq 2000. While differences in error rates exist between Illumina instruments, no specific assessment has been made between these NextSeq models [42]. However, the absolute differences in error rates are minor and unlikely to impact our study significantly. This is particularly relevant since Illumina's lower F1 score compared to ONT was due to missed calls rather than erroneous ones.

In summary, while there may be specific equipment or preparation artifacts to consider, we took steps to minimise these effects and maintain consistency across our sequencing methods.

**Reviewer #3 (Public Review):**
Hall et al. benchmarked different variant calling methods on Nanopore reads of bacterial samples and compared the performance of Nanopore to short reads produced with Illumina sequencing. To establish a common ground for comparison, the authors first generated a variant truth set for each sample and then projected this set to the reference sequence of the sample to obtain a mutated reference. Subsequently, Hall et al. called SNPs and small indels using commonly used deep learning and conventional variant callers and compared the precision and accuracy from reads produced with simplex and duplex Nanopore sequencing to Illumina data. The authors did not investigate large structural variation, which is a major limitation of the current manuscript. It will be very interesting to see a follow-up study covering this much more challenging type of variation.

We fully agree that investigating structural variations (SVs) would be a very interesting and important follow-up. Identifying and generating ground truth SVs is a nontrivial task and we feel it deserves its own space and study. We hope to explore this in the future.

In their comprehensive comparison of SNPs and small indels, the authors observed superior performance of deep learning over conventional variant callers when Nanopore reads were basecalled with the most accurate (but also computationally very expensive) model, even exceeding Illumina in some cases. Not surprisingly, Nanopore underperformed compared to Illumina when basecalled with the fastest (but computationally much less demanding) method with the lowest accuracy. The authors then investigated the surprisingly higher performance of Nanopore data in some cases and identified lower recall with Illumina short read data, particularly from repetitive regions and regions with high variant density, as the driver. Combining the most accurate Nanopore basecalling method with a deep learning variant caller resulted in low error rates in homopolymer regions, similar to Illumina data. This is remarkable, as homopolymer regions are (or, were) traditionally challenging for Nanopore sequencing.Lastly, Hall et al. provided useful information on the required Nanopore read depth, which is surprisingly low, and the computational resources for variant calling with deep learning callers. With that, the authors established a new state-of-the-art for Nanopore-only variant, calling on bacterial sequencing data. Most likely these findings will be transferred to other organisms as well or at least provide a proof-of-concept that can be built upon.As the authors mention multiple times throughout the manuscript, Nanopore can provide sequencing data in nearly real-time and in remote regions, therefore opening up a ton of new possibilities, for example for infectious disease surveillance.However, the high-performing variant calling method as established in this study requires the computationally very expensive sup and/or duplex Nanopore basecalling, whereas the least computationally demanding method underperforms. Here, the manuscript would greatly benefit from extending the last section on computational requirements, as the authors determine the resources for the variant calling but do not cover the entire picture. This could even be misleading for less experienced researchers who want to perform bacterial sequencing at high performance but with low resources. The authors mention it in the discussion but do not make clear enough that the described computational resources are probably largely insufficient to perform the high-accuracy basecalling required.

We have provided runtime benchmarks for basecalling in Supplementary Figure S23 and detailed these times in Supplementary Table S7. In addition, we state in the Results section (P9 L239-241) “Though we do note that if the person performing the variant calling has received the raw (pod5) ONT data, basecalling also needs to be accounted for, as depending on how much sequencing was done, this step can also be resource-intensive.”

Even with super-accuracy basecalling considered, our analysis shows that variant calling remains the most resource-intensive step for Clair3, DeepVariant, FreeBayes, Medaka, and NanoCaller. Therefore, the statement “the described computational resources are probably largely insufficient to perform the high-accuracy basecalling required”, is incorrect. However, we have made this more prominent in the Results and Discussion.

In the results section we added the underlined section:

“… FreeBayes had the largest runtime variation, with a maximum of 597s/Mbp, equating to 2.75 days for the same genome. In contrast, basecalling with a single GPU using the super-accuracy model required a median runtime of 0.77s/Mbp, or just over 5 minutes for a 4Mbp genome at 100x depth. …”

In the discussion we have added the following statement:

“Basecalling is generally faster than variant calling, assuming GPU access, which is likely considered when acquiring ONT-related equipment.”

**Recommendations for the authors:**

**Reviewer #1 (Recommendations For The Authors):**
The colour choices in Figure 3 and Figure 4 c made the illustrations somewhat difficult to read. More substantially, a deeper investigation of the causes of non-homopolymeric-related mistaken indel calls would be useful.

We have updated Figure 3 so that each line has a different style to aid in discriminating between colours. The colour scheme for Figure 4c has also been updated.

In terms of non-homopolymeric false positive (FP) indel calls, we did an investigation of these for Clair3 and DeepVariant on the simplex sup data as these are the two best performing variant callers and deal the best with homopolymers. For Clair3, there were eight FPs across all samples. Five of these were homopolymers. The remaining three occurred within one or two bases of another insertion which inserted a similar sequence to the FP. For DeepVariant, it was much the same story, with 8/11 FP indels being in homopolymers, and the remaining three being within one or two bases of another insertion with a similar sequence. We have added a couple of sentences to the results explaining this finding.

**Reviewer #2 (Recommendations For The Authors):**
The paper is well-written and provides evidence for the conclusions. Some issues should be addressed.Include a section in the Results describing species-specific observations, namely if some samples had recurrently lower SNP and INDEL F1 scores (as observed in Figure 2).

Please see our response in your second point in the ‘Weaknesses’ section of the public review.

Please provide more details on how the samples were sequenced. Section "Sequencing" in the methods is confusing and not clear enough to be reproduced (provide a supplementary table/figure with the workflow for each sample). Add information about how many samples were multiplexed in each run and what was the output achieved in each.

We have now added a Supplementary Table S9 which outlines which instruments, kits, and multiplexing strategies were used for each sample. In addition, the raw pod5 data that we make available has been segregated by sample, so knowledge of the multiplexing strategy is not necessary for someone attempting to reproduce our results.

The authors acknowledge that structural variation was not evaluated in this manuscript. Since ONT sequencing is often used to reconstruct the sequence of plasmids for outbreak/epidemiology analysis, perhaps they could undertake this analysis on a plasmids dataset (which suffers from constant structural variation).

As noted in our response to Reviewer 3’s public review, we fully agree that investigating structural variations (SVs) would be a very interesting and important follow-up. Identifying and generating ground truth SVs is a nontrivial task and we feel it deserves its own space and study. We hope to explore this in the future.

Reviewer #3 (Recommendations For The Authors):The manuscript is well organized. However, some sections are a bit long and would benefit from being more concise.

Thank you for your valuable feedback and for acknowledging the organisation of our manuscript. We appreciate your suggestion regarding the length of certain sections. We have gone back through and made the manuscript more concise.

Figure 1: Is the Qscore really the same as identity? Isn't the determination of identity only possible after alignment?

When we say Qscore we are referring to the Phred-scaled version of the read identity, which is alignment based, not the Qscores of the individual bases in the FASTQ file. We have updated the text and figure legend to make this clearer. “The Qscore is the logarithmic transformation of the read identity, , where 𝑃 is the read identity.”. We also now explicitly state that read identity is alignment-based.

Abbreviations/terms mentioned but not introduced:- kmers, P2L57- ANI, P3L93

We have updated the text to better introduce these terms.